# Integrative Roles of Dopamine Pathway and Calcium Channels Reveal a Link between Schizophrenia and Opioid Use Disorder

**DOI:** 10.3390/ijms24044088

**Published:** 2023-02-17

**Authors:** Siroshini K. Thiagarajan, Siew Ying Mok, Satoshi Ogawa, Ishwar S. Parhar, Pek Yee Tang

**Affiliations:** 1Department of Mechatronics and Biomedical Engineering, Universiti Tunku Abdul Rahman, Kajang 43000, Malaysia; 2Jeffrey Cheah School of Medicine and Health Sciences, Monash University Malaysia, Subang Jaya 47500, Malaysia

**Keywords:** addiction, behavior, calcium channel, domperidone, dopamine, morphine, schizophrenia, zebrafish

## Abstract

Several theories have been proposed to explain the mechanisms of substance use in schizophrenia. Brain neurons pose a potential to provide novel insights into the association between opioid addiction, withdrawal, and schizophrenia. Thus, we exposed zebrafish larvae at 2 days post-fertilization (dpf) to domperidone (DPM) and morphine, followed by morphine withdrawal. Drug-induced locomotion and social preference were assessed, while the level of dopamine and the number of dopaminergic neurons were quantified. In the brain tissue, the expression levels of genes associated with schizophrenia were measured. The effects of DMP and morphine were compared to vehicle control and MK-801, a positive control to mimic schizophrenia. Gene expression analysis revealed that *α1C*, *α1Sa*, *α1Aa*, *drd2a*, and th1 were up-regulated after 10 days of exposure to DMP and morphine, while *th2* was down-regulated. These two drugs also increased the number of positive dopaminergic neurons and the total dopamine level but reduced the locomotion and social preference. The termination of morphine exposure led to the up-regulation of *th2*, *drd2a*, and *c-fos* during the withdrawal phase. Our integrated data implicate that the dopamine system plays a key role in the deficits in social behavior and locomotion that are common in the schizophrenia-like symptoms and opioid dependence.

## 1. Introduction

Substance use disorders and schizophrenia often co-occur. According to epidemiological studies, patients with schizophrenia have serious problems with alcohol dependence, smoking, and illicit drug dependence [1,2,3]. Addiction to several drug classes, including cannabis, methamphetamine, and cocaine, occurs commonly in patients with schizophrenia [2]. The prevalence of these substance use disorders is significantly elevated among schizophrenics compared to individuals with other psychiatric diagnoses or the general population [4,5,6]. Patients with first episode psychosis were more likely to be cannabis users [7], and therefore adolescent cannabis might increase the risk of schizophrenia [8,9,10]. Moreover, illicit drugs worsen schizophrenia symptoms, limit treatment compliance, increase psychotic relapse and hospitalization, and increase the risks of readmission and suicide [11,12,13,14,15,16,17]. There are some explanations for the link between schizophrenia and addiction, and one of them suggests that these two disorders are influenced by a common biological trait.

Dopaminergic dysfunction is thought to play a role in the biological hypotheses of both schizophrenia and addiction. It was suggested that dopaminergic hyperactivity could be responsible for the positive symptoms of schizophrenia [18]. Patients with schizophrenia show increases in subcortical synaptic dopamine content [19,20] and increased basal dopamine synthesis capacity [21,22,23]. On the contrary, the negative symptoms are associated with dopaminergic deficiency in the prefrontal cortex [18,24]. Due to the heterogeneity of negative symptoms, certain symptoms such as motivation impairment, underlying avolition, asociality, and anhedonia can derive from specific dopamine (DA) pathophysiological mechanisms [25]. Dopaminergic innervations from the ventral tegmental area to the basolateral amygdala are involved in emotional processing and associative learning of rewarding and aversive stimuli. An anatomical difference in these projections between males and females indicates sex differences in motivational and emotional behaviors and related psychiatric dysfunctions [26]. Alterations of the DA-dependent response of the ventral striatum to reward anticipation could be the cause of these motivational deficits, which are associated with a reduced sensitivity to reinforcers and pleasure [25,27]. Elevated striatal DA levels, which may be important in schizophrenia, are attributable to deficits in glutamate synapses and synaptic plasticity [28].

Addiction is a dysregulation by a pathological imbalance of the brain’s reward systems. Most of the drugs of abuse induce the increase of extracellular DA concentration in the limbic regions, including the nucleus accumbens (NAc) [29,30], via interaction of different DA receptors [31]. These psychoactive drugs include stimulants [32,33], nicotine [34], alcohol [35], and marijuana [36]. Increases in DA levels are crucial in coding, predicting, and motivating the acquisition of reward [37]. Through conditioning of motivation of reward procurement, a neutral stimulus which is linked with the reinforcer, activates rapid DA release in the striatum, encoding reward-directed behaviors [38].

Various clusters of dopaminergic neurons with different anatomical positions and functions are localized in the diencephalon, mesencephalon, and the olfactory bulb [39]. DA is synthesized in the neurite of dopaminergic neurons, regulated by a set of enzymes including tyrosine hydroxylase (TH) and dopa decarboxylase (DDC). The amino acid tyrosine is converted into 3,4-dihydroxyphenylalanine (DOPA), the precursor of DA by TH, a rate-limiting enzyme in DA biosynthesis. Following that, DDC catalyzes the irreversible decarboxylation of DOPA, followed by the production of DA [40,41]. Finally, DA is released at the nerve terminals to activate post-synaptic DA (D1- or D2-type) receptors, and presynaptic D2-type autoreceptors. D1- and D2-type receptors are linked to stimulation and inhibition of adenylate cyclase respectively, where the latter is involved in the regulation of DA synthesis, metabolism, and release [42,43].

The firing of an action potential in response to depolarization of neurons relies on a number of different types of voltage-gated ion channels that are permeable to sodium, potassium, chloride, and calcium. The former three ions support a predominantly electrogenic role, while calcium ions alter membrane potentials and serve as important signaling entities [44]. Voltage-gated calcium channels (CaVs) are categorized into two major groups. High voltage-activated (HVA) channels are activated by large membrane depolarizations, while low voltage-activated (LVA) channels are triggered by smaller voltage changes [45,46]. L-type calcium channels (LTCCs or Ca_v_1), members of the high-voltage activated family, are widely expressed in the central nervous system [47]. LTCCs have been suggested to mediate DA D2 receptors (D2R) responses [48]. Recently, genes encoding CaVs have been implicated in psychiatric disorders [49,50,51]. Notably, *cacna1* gene encoding CaV subunit alpha 1, the pore-forming α_1_ subunit, has been associated with schizophrenia, as elucidated by previous genetic and biological studies [52,53,54,55,56]. However, the role of these CaV-encoding genes in the pathophysiology of schizophrenia remains unexplored.

Zebrafish (*Danio rerio*) are fast-emerging as a powerful model to study psychiatric diseases due to its numerous advantages [57]. Zebrafish possess about 70% gene homology to humans [58] and share similar brain architecture with humans [59]. Furthermore, neurotransmitter systems are conserved between zebrafish and mammals, allowing for the translation of neurotransmission changes and associated developmental and disease pathways [60]. The dopaminergic system in zebrafish, which is fully formed by 96 h post-fertilization (hpf), has been widely studied [61]. Another important advantage is their high fecundity, which makes them easy for high-throughput drug screening [57]. Lastly, their transparency at early stages of development allows for non-invasive in vivo imaging using transgenic reporter lines.

Considering drug abuse and negative symptoms in schizophrenia are largely related to the DA activity in the ventral striatum [30,62,63,64,65], it is suggested that there is a possible link between addiction and schizophrenia. Based on the considerable comorbidity between schizophrenia and drug abuse, we were interested in using a zebrafish model to elucidate the differential effects of morphine and domperidone (DMP), a D2R antagonist on dopaminergic neurons leading to alterations in gene expression, dopamine synthesis, locomotor, and social behaviors.

## 2. Results

The short-term exposure to MK-801 significantly enhanced their locomotion activity. In contrast to MK-801, DMP and morphine reduced swimming activity (Figure 1A). When the larvae were exposed to these three drugs for 10 days, we observed a depression of locomotor activity (Figure 1B). Increased locomotor activity in zebrafish larvae in response to morphine withdrawal was observed after the termination of short- and long-term morphine exposure (Figure 1C,D). Apart from the difference in locomotor activity, all treated groups demonstrated a different travelling distance from the control group. When comparing with locomotor activity (Figure 1A,B), the short- and long-term treated larvae showed a similar trend in travelling distance (Figure 2A,B). However, the morphine withdrawal groups displayed a different swimming pattern. Although they crossed the central line with higher frequency (Figure 1C,D), they merely swam close to the central line, resulting in low travelling distance (Figure 2C,D).

While the controls exhibited a strong, positive, assortative preference toward conspecifics, the other three treated groups, by contrast, showed significantly decreased sociability (Figure 3A,B). During the morphine withdrawal periods, the larvae still did not preferentially associate with conspecifics (Figure 3C,D). The larvae from all the treated groups displayed different degrees of avoidance to conspecifics (Figure 4).

Dopaminergic neurons were seen distributed in brain regions mainly in the periventricular nucleus of posterior tubercule (TPp), posterior tuberculum (PT), caudal hypothalamus (Hc), and locus coeruleus (LC). A noticeable increase was seen in the distribution of dopaminergic neurons, especially in the PT region, between the control and the drug-treated larvae (Figure 5A,B). Exposure to drugs, particularly MK801, was associated with a significant increment of dopaminergic cell number (Figure 6A,B). Nonetheless, during the morphine withdrawal, there was no significant difference in the number of dopaminergic neurons between the control and the morphine treated group (Figure 5C,D and Figure 6C,D). Histograms (Figure 7A,B) show that all treatments increased concentration of DA, reflecting increment of dopaminergic cell number. However, DA levels remained high during morphine withdrawal (Figure 7C,D).

We found that *eif4bp1* was up-regulated by all short-term drug treatments. In contrast, *α1Sa*, *drd2a*, and *th2* were down-regulated. Of the three drug treatments, only morphine down-regulated *th1* (Figure 8A). After long-term drug treatments, the gene expression changed more dramatically as compared to short-term treatments. *α1C*, *α1Sa*, *α1Aa*, *drd2a*, and *th1* were up-regulated by all three drugs. Among these genes, MK-801 gave the most prominent effect to *α1C*, *α1Sa*, and *α1Aa*, while DMP and morphine extensively up-regulated *drd2a* and *th1*. *C-fos* was up-regulated by MK-801 and DMP, whereas *eif4bp1* was up-regulated by both MK-801 and morphine. *Th2* was the only gene down-regulated after long-term drug treatments (Figure 8B). When the larvae underwent morphine withdrawal after short-term morphine exposure, *c-fos*, *th2*, and *drd2a* were down-regulated while *th1* and *eifbbp1* were up-regulated (Figure 8C). After long-term exposure to morphine, *α1Aa*, *c-fos*, *drd2a, th1*, and *eif4bp1* remained up-regulated during withdrawal. Surprisingly, the expression of *th2* was highly up-regulated (Figure 8D).

## 3. Discussion

DA neurons are characterized by the expression of D2R [66]. Findings suggest elevated density of D2R in the brains of schizophrenia patients [19,67]. The encoding gene, *drd2* expression, was found to be related with negative symptoms and positively correlated with the deficit syndrome severity [68]. There are two *drd2* orthologs in zebrafish, *drd2a* and *drd2b*, due to the whole genome duplication in teleosts [69], which show a similarity of 27% and 66%, respectively, with human *drd2* [70]. In the current study, all short-term drug exposures led to suppression of *drd2a*. However, the long-term exposure to these drugs induced upregulation of *drd2a*. NMDA antagonists such as MK-801 and 3-(2-carboxypiperazin-4-yl)-propyl-1-phosphonic acid, a competitive NMDA receptor antagonist, have been reported to upregulate postsynaptic *drd2* [71,72,73,74]. These antagonists block the presynaptic NMDA receptors on the dopaminergic projections [75,76] and inhibit the release of DA, which results in an upregulation of postsynaptic *drd2*. Our observations regarding *drd2a* elevation in long-term morphine treated and morphine withdrawn larvae further support the previous finding regarding the association of increased *drd2* to morphine dependence and withdrawal [77]. In the present study, confocal microscopy analysis revealed that the elevation of *drd2a* could be explained by the increased number of DA neurons.

TH is a rate-limiting enzyme which catalyzes the transformation of L-tyrosine to the dopamine precursor (L-DOPA). This enzyme has been widely used as a molecular marker of dopaminergic neurons. In zebrafish, TH is coded by *th1* and *th2* [78,79]. We found that the expression of *th1* after long-term drug treatments was comparable with the increase of dopaminergic neurons. The increase of *th1* was reflected by the number of DA neurons, and this was further verified by the raise of DA level. In contrast, the expression of *th2* was significantly reduced in this study. Unlike *th1*, which is widely expressed in various brain regions [79], the expression of *th2* is restricted to the preoptic nucleus and the cerebrospinal fluid-contacting neurons in the PT, and the intermediate and caudal hypothalamus, where there is little co-expression of *th1* [79,80]. These *th2*-expressing neurons are involved in the maintenance of normal levels of spontaneous activity and production of swimming behaviors [81,82]. Our data support the reduction of *th2* expression could be associated with the low number of movements after short- and long-term exposure to MK-801, DMP and morphine. The highly expressed *th2* in the larvae during morphine withdrawal after long-term morphine exposure might be reflected by their hyperactive swimming behavior.

As an immediate early gene, *c-fos* serves as a marker of neuronal activation [83]. It has been reported that MK-801 increased *c-fos* expression in the schizophrenia rat model [84]. The MK-801-induced *c-fos* expression in the long-term treatment group is in line with previous findings. Surprisingly, morphine failed to induce *c-fos* expression after long-term exposure. An increase of *c-fos* expression was observed only after termination of long-term morphine treatment. Previous studies reported that morphine does not induce FOS [85,86]. However, opiate withdrawal increases the expression of the *c-fos* in different brain regions, such as the NAc, amygdala, and GABAergic tail of the ventral tegmental area [87,88,89,90]. Although *c-fos* mRNA in zebrafish has been evaluated following various experimental manipulations [91], the effect of D2R antagonist on *c-fos* expression has not been investigated in the larval brain. The blockade of D2R in the prefrontal cortex was reported to induce c-FOS expression in the dorsomedial striatum, dorsolateral striatum, NAc shell, lateral septal nucleus ventral part, and bed nucleus of the stria terminalis in mice [92]. This could explain the significant upregulation of *c-fos* after long-term DMP treatment.

Voltage-gated calcium channels (Ca_v_s) are transmembrane proteins activated by depolarization of membrane potential. Classically, dysfunction of Ca_v_s has been linked to neurological disorders including Parkinson’s disease, ataxia, migraine, and neuropathic pain [93]. Ca_v_s are being considered as molecular targets to treat several neurological conditions including psychiatric disorders [50]. Ca_v_1 channel subfamily comprises Ca_v_1.1. Ca_v_1.2, Ca_v_1.2, and Ca_v_1.4 channels, while the Ca_v_2 channel subfamily comprises Ca_v_2.1, Ca_v_2.2, and Ca_v_2.3. These channels are involved in the release of neurotransmitters. Calcium ion influx via Ca_v_1 postsynaptic channels activates phosphorylation of cyclic-AMP response element binding protein (CREB) [94,95,96,97] and NFATc4 neuronal nuclear transcription factors [98]. Ca_v_2.1 and Ca_v_2.2 channels play a vital role in the release of fast transmitters such as GABA, acetylcholine, and glutamate [99,100]. In the current study, genes encoding Ca_v_1.1, Ca_v_1.2, and Ca_v_2.1 were up-regulated by long-term drug treatments. The up-regulation of these Ca_v_s will lead to a sustained activation of CREB in the NAc, which leads to anhedonia-like and pro-depression-like symptoms [101,102]. This could be further supported by our previous study, where *creb1* was up-regulated after treatment with DMP and morphine [103]. The upregulation of these Ca_v_s will also allow excessive influx of calcium ions inside the cells. The overloaded calcium will reduce the amount of ATP production while increasing the accumulation of reactive oxygen species and release of cytochrome C that induces apoptosis of neuronal cells [104]. Excessive accumulation of calcium ions in the mitochondria of the neurons can also lead to excessive neuronal firing and eventually cause neuronal death. The dysfunction of mitochondria can lead to several diseases, including intellectual disability [105]. Moreover, a gain-of-function mutation in Ca_v_1.2 causes Timothy syndrome [106], which is associated with neurological developmental defects, including manifestation of neuropsychiatric phenotypes [106,107,108]. This can be reflected by the lower social preference in all the MK-801, DMP, and morphine-treated larvae. Noteworthily, for the morphine withdrawn larvae, low social preference was still shown, and this could be due to the high expression of *α1Aa* which encodes Ca_v_2.1.

The eukaryotic translation initiation factor 4E binding protein 1 (EIF4EBP1), encoded by *eif4ebp1*, is one member of a family of small proteins that act as repressors of mRNA translation initiation by binding the mRNA cap-binding protein eIF4E [109]. The activity of phosphatidylinositol 3-kinase (PI3K)/AKT pathway negatively regulates the transcription of EIF4EBP1 via mechanistic target of rapamycin (mTOR) [110]. The significance of mTOR in neuron development and its regulation of translational control and protein synthesis implicated that mTOR is involved in the pathology of schizophrenia [111,112,113,114]. Although the role of mTOR was not tested in the current study, upregulation of *eif4ebp1* after drug exposure and morphine withdrawal could be explained by the downregulation of *pi3k* and *akt1* in our previous study [103].

## 4. Materials and Methods

### 4.1. Zebrafish Strains and Housing Conditions

Wild-type zebrafish embryos at 0 hpf were collected from Danio Assay Laboratories Pvt. Ltd., University Putra Malaysia, Serdang, Selangor, Malaysia. Transgenic Tg (*dat*:EGFP) zebrafish embryos were obtained from Jeffrey Cheah School of Medicine and Health Sciences, Monash University Malaysia, Subang Jaya, Selangor, Malaysia, after 2 hpf. This strain was developed by the Center for Advanced Research in Environmental Genomics, Department of Biology, University of Ottawa. The embryos were maintained at 27 °C in embryo medium (5 mM sodium chloride (NaCl), 0.17 mM potassium chloride (KCl), 0.33 mM calcium chloride (CaCl_2_), and 0.33 mM magnesium sulfate (MgSO_4_), pH 7.4). Unfertilized, unhealthy, and dead embryos were discarded. The hatched larvae were fed at 7 days post-fertilization (dpf) with live paramecium. All experiments were done with the approval of animal ethics by Universiti Tunku Abdul Rahman Research Ethics and Code of Conduct (U/SERC/18/2020). Visual screening was performed within the time frame of experimental procedure to monitor the health of the zebrafish larvae by removing the larvae with abnormal mortality, heartbeat, and swimming activity.

### 4.2. Drug Optimization and Treatments

The optimization of maximum-tolerated concentration (MTC) of each drug and identification of its optimal condition in inducing schizophrenia were based on our previous study [103]. At 2 dpf, both strains of the zebrafish were dosed at MTC according to the short- (3 days) and long-term (10 days) treatment protocols. DMP and morphine were added to the embryo medium at a final concentration of 3.13 μM and 0.80 μM, respectively. Additionally, 5.0 μM of dizocilpine (MK-801) (Sigma-Aldrich, Saint Louis, Missouri, USA), a schizophrenia-mimicking compound, was used as a positive control, while embryo medium served as vehicle control. Three groups (n = 50 /group) of embryos were examined for the effect of short- and long-term treatments. Another two groups of embryos were assessed for withdrawal syndrome five days after the termination of short- and long-term morphine treatments.

### 4.3. Behavioral Analyses

Calculating the total number of crossings is a simple and effective method to determine the locomotor activity of zebrafish [115,116]. Our assay was modified from the method of Boehmler et al. (2007) [115]. A single vertical line was drawn on the bottom of a clear petri dish (60 mm × 15 mm) which divided the dish into 2 equal halves. Three larvae were placed in the petri dish filled with embryo medium at 27 °C. Locomotor activity was measured for 30 s by counting the number of times each larva crossed the line. Total distance traveled was also used as a measure of locomotor activity. A recording camera at 30 frames per second was positioned above the arena. Videos were captured for 5 min and processed using ToxTrac [117]. Each treatment was subjected to 12 independent trials with 3 independent experiments.

### 4.4. Social Preference

The experimental tank (14 cm length, 7 cm width, 5 cm height) was separated by a transparent divider into exposure compartment and conspecific compartment (Figure A1). To avoid lateral bias in zebrafish cohorts, the left/right location of target/conspecific larvae were alternated between the trials. The target larvae were pre-exposed to a drug or drug-free water (control) for 20 min. Control or drug-exposed zebrafish (n = 12 in each group) were introduced individually to the central zone of the apparatus, temporarily separated (by transparent sliding dividing doors) from the two arms of the corridor.

The larva was subjected to an initial 30-s acclimation interval in order to reduce transfer/handling stress. The larva was allowed to explore the apparatus for 5 min. Videos were captured to record the time of each larva spent near the conspecific compartment.

### 4.5. Gene Expression Analysis by Real-Time PCR

The larvae were euthanized by rapid cooling in ice water (0–4 °C) for at least 20 min [118]. Total RNA from the dissected brains of zebrafish larvae pool (n = 50) was extracted using TRIzol reagent (Thermo Fisher Scientific, Waltham, Massachusetts, USA), with three independent pools used for each treatment [119]. The quality of RNA was assessed by gel electrophoresis and the total RNA concentration was determined using spectrophotometer. The double-stranded cDNA was synthesized using a Tetro cDNA Synthesis Kit (Bioline, London, UK). Eight genes related to the dopaminergic circuitry were selected. The primers used to amplify these genes are presented in Table 1. Quantitative real-time PCR was performed using the SensiFAST™ SYBR^®^ No-ROX kit (Bioline, London, UK) in a qTOWER3 G real-time thermal cycler (Analytik Jena, Jena, Germany) with beta actin (β-actin) as the housekeeping gene. cDNA was diluted in series ranging from 100 ng/μL to 0.01 ng/μL to generate a standard curve. All reactions were performed in triplicate. The relative expression of each gene was calculated from the average CT (cycle threshold) readings at each point of the standard curve using the formula E = 10 (1/slope). Expression of genes of interest was then normalized to the housekeeping gene.

### 4.6. Total Protein Purification

Zebrafish larvae were washed with cold phosphate buffered saline (PBS) and stored overnight at −20 °C. Cell lysis was conducted by two freeze–thaw cycles. The pooled larval brain tissue was homogenized in PBS containing 1% Triton-X and sonicated for 10 min. The homogenates were centrifuged at 5000× *g* for 5 min. The supernatant was collected, and subsequently the protein concentration was determined using Bradford assay. The purified protein was subjected to biochemical assays immediately.

### 4.7. Biochemical Assay

The concentration of DA was determined using the commercial enzyme-linked immunosorbent assay (ELISA) kits (Cusabio Biotech^®^, Houston, Texas, USA) according to the manufacturer’s instructions. The optical density (OD value) of each well was determined at 450 nm for the assay.

### 4.8. *Confocal Microscopy*

Tg (*dat*:EGFP) larvae were treated with 1-phenyl-2-thiourea (0.003% phenylthiourea, Sigma-Aldrich, Saint Louis, Missouri, USA) to prevent pigment formation. Preceding the treatments, the larvae were fixed by 4% paraformaldehyde (PFA) overnight at 4 °C and stored in PBS until analysis. Prior to imaging, each larva was embedded in 2% low melting point agarose gel (Sigma, Darmstadt, Germany). A small incision was made to allow the dorsal of the larvae to face down in order to image the brain region. The brain image of the larva was then captured by the laser scanning confocal microscope (Nikon C1si, Nikon Inc., Tokyo, Japan) and the NIS Elements AR software version 4.1 (Nikon Inc., Tokyo, Japan). The dopaminergic neurons in the PT were determined based on the z-stacks of confocal images using the ImageJ (Bethesda, Maryland, USA) 3D object counter plugin described by Tay et al. [127].

### 4.9. Statistical Analysis

All results were expressed as mean ± SEM. The differences between samples were analyzed using the one-way analysis of variance (ANOVA) followed by post-hoc Tukey’s test. Means comparisons between the control and withdrawal groups were performed using Student’s *t*-test. All statistical tests were performed using Statistical Program for Social Sciences Statistical Program for Social Sciences (SPSS version 22). A *p*-value of less than 0.05 (*p* < 0.05) was considered statistically significant.

## 5. Conclusions

In summary, our study, which integrated behavioral, biochemical, gene expression, and confocal microscopy imaging data in drug-treated zebrafish larvae, revealed that there is an association between opioid addiction and schizophrenia. The alterations in dopaminergic neurons, calcium channels, and gene expression induced by DMP and morphine could contribute to the abnormal locomotor and social behaviors. Future work is required to elucidate the specificity of these changes in the context of these two disorders.

## Figures and Tables

**Figure 1 ijms-24-04088-f001:**
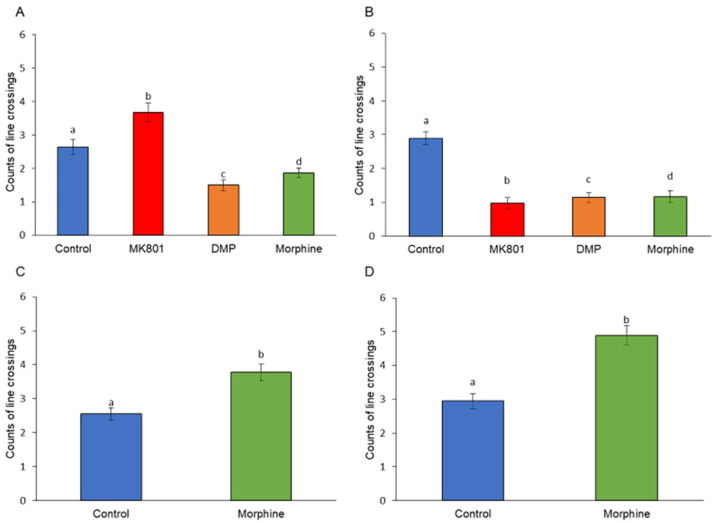
Effects of (**A**) short-term (F (3, 140) = 42.92, *p* = 0.01) and (**B**) long-term (F (3, 140) = 114.29, *p* = 0.00) exposure to MK801, DMP, and morphine, and after withdrawal of (**C**) short-term (t (70) = 35.30, *p* = 0.00) and (**D**) long-term (t (70) = 177.64, *p* = 0.00) morphine treatment on the locomotor activity. Data expressed as mean ± SEM, *n* = 12 of 3 independent experiments. Different alphabets indicate statistically significant values, *p* < 0.05. Drug abbreviation: DMP, domperidone.

**Figure 2 ijms-24-04088-f002:**
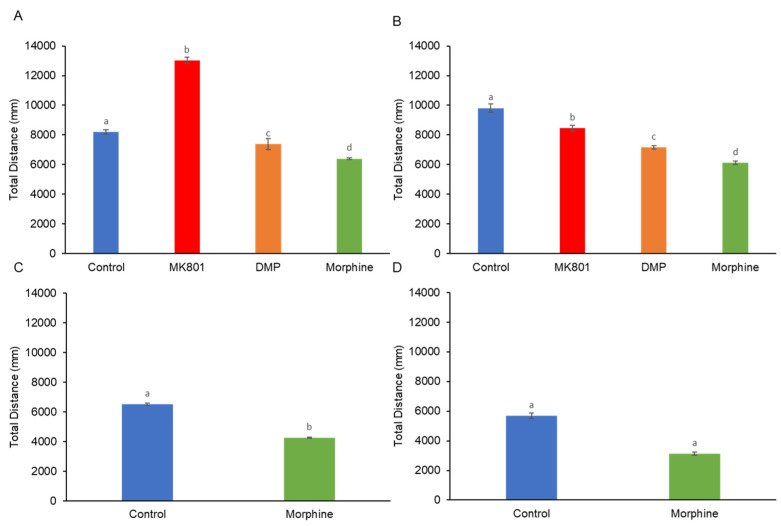
Effects of (**A**) short-term (F (3, 140) =46.84, *p* = 0.01) and (**B**) long-term (F (3, 140) = 124.51, *p* = 0.01) exposure to MK801, DMP, and morphine, and after withdrawal of (**C**) short-term (t (70) = 32.81, *p* = 0.04) and (**D**) long-term (t (70) = 184.32, *p* = 0.01) morphine treatment on the total distance travelled of zebrafish larvae (mm/5 min). Data expressed as mean ± SEM, *n* = 12 of 3 independent experiments. Different alphabets indicate statistically significant values, *p* < 0.05. Drug abbreviation: DMP, domperidone.

**Figure 3 ijms-24-04088-f003:**
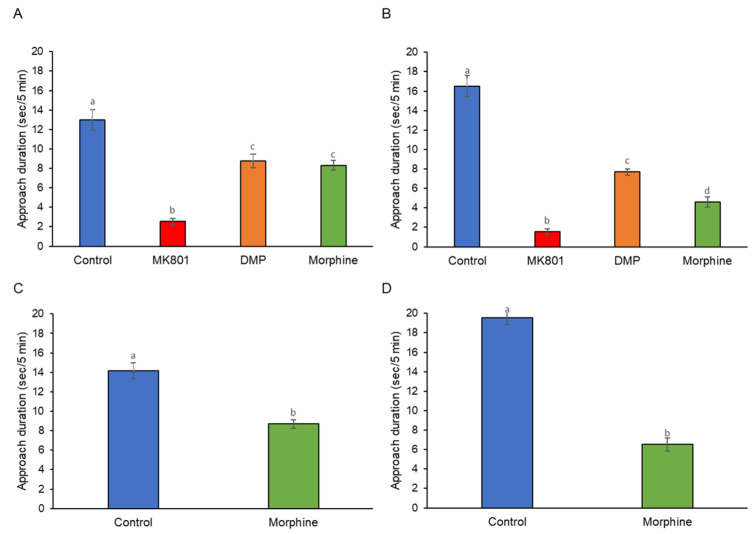
Effects of (**A**) short-term (F (3, 140) =42.92, *p* = 0.00) and (**B**) long-term (F (3, 140) = 114.29, *p* = 0.00) exposure to MK801, DMP, and morphine, and after withdrawal of (**C**) short-term (t (70) = 35.31, *p* = 0.03) and (**D**) long-term (t (70) = 177.64, *p* = 0.00) morphine treatment on the social preferences of zebrafish larvae (sec/5 min). Data expressed as mean ± SEM, *n* = 12 of 3 independent experiments. Different alphabets indicate statistically significant values, *p* < 0.05. Drug abbreviation: DMP, domperidone.

**Figure 4 ijms-24-04088-f004:**
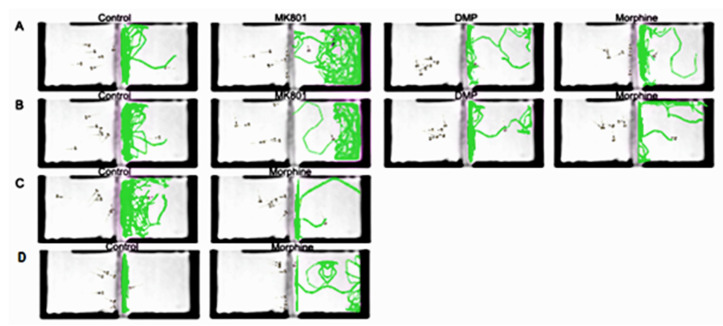
Tracked trajectories for control, MK801, DMP and morphine treated larvae after being subjected to conspecific larvae. Effects of (**A**) short-term and (**B**) long-term exposure to MK801, DMP and morphine, and after withdrawal of (**C**) short-term and (**D**) long-term morphine treatment on the social preference. Drug abbreviation: DMP, domperidone.

**Figure 5 ijms-24-04088-f005:**
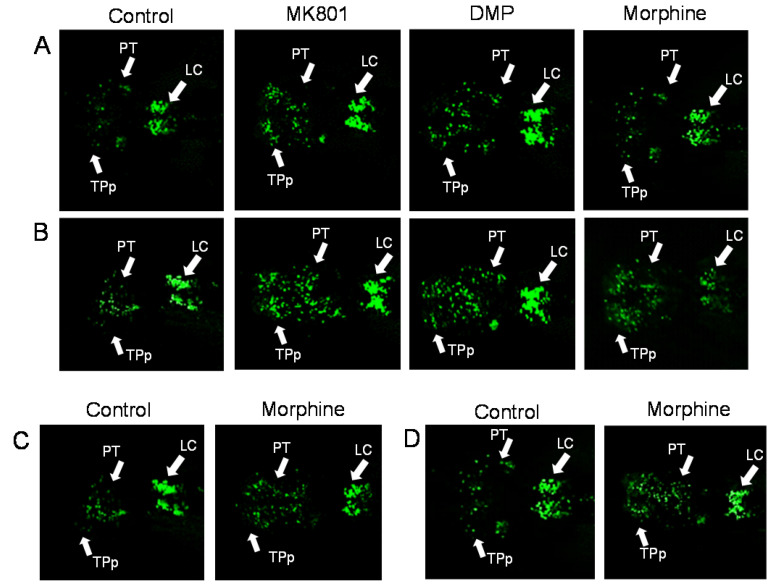
Distribution of dopaminergic neurons in Tg (*dat*: EGFP) zebrafish. Representative confocal images showing the location of dopaminergic nuclei in the brains of the larval zebrafish following (**A**) short-term and (**B**) long-term exposure to MK801, DMP, and morphine, and after withdrawal of (**C**) short-term and (**D**) long-term treatment of morphine. Drug abbreviation: DMP, domperidone. Other abbreviations: LC, locus coeruleus; PT, posterior tuberculum; TPp, posterior tubercule.

**Figure 6 ijms-24-04088-f006:**
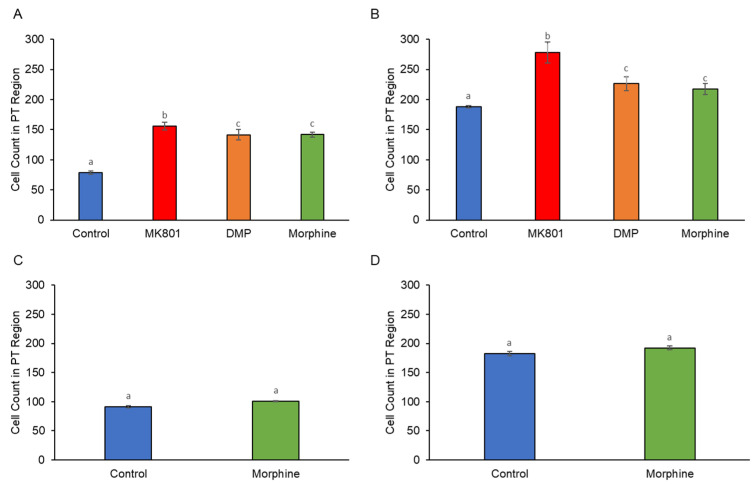
Effects of (**A**) short-term (F (3, 28) = 19.32, *p* = 0.00), (**B**) long-term (F (3, 28) = 10.74, *p* = 0.00) exposure to MK801, DMP, and morphine, and after withdrawal of (**C**) short-term (t (14) = 18.78, *p* = 0.00) and (**D**) long-term (t (14) = 6.33, *p* = 0.02) morphine treatment on cell counts in the PT region of the zebrafish larvae brain. Data expressed as mean ± SEM, *n* = 8. Different alphabets indicate statistically significant values, *p* < 0.05. Drug abbreviation: DMP, domperidone. Other abbreviation: PT, posterior tuberculum.

**Figure 7 ijms-24-04088-f007:**
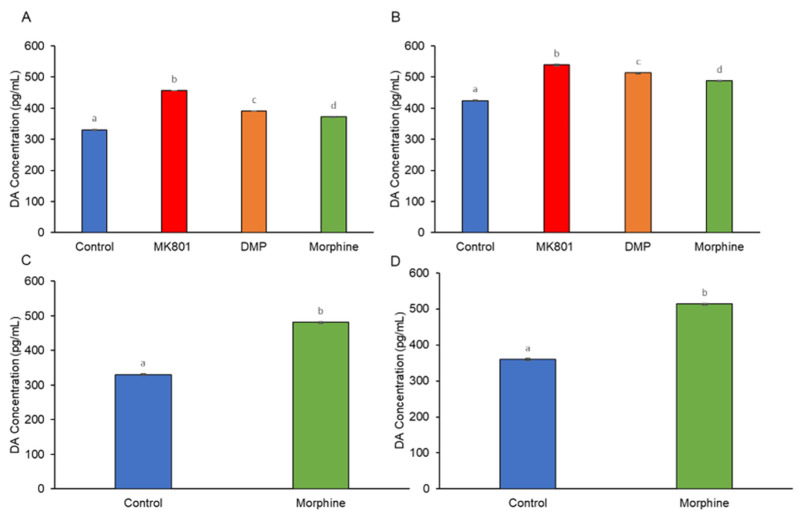
Effects of (**A**) short-term (F (3, 32) = 20.09, *p* = 0.00) and (**B**) long-term (F (3, 32) = 22.17, *p* = 0.00) exposure to MK801, DMP, and morphine, and after withdrawal of (**C**) short-term (t (16) = 206.7, *p* = 0.00) and (**D**) long-term (t (16) = 56.83, *p* = 0.00) morphine treatment on the DA concentration (pg/mL) of the zebrafish larvae brain. Data expressed as mean ± SEM, *n* = 50 of 3 independent experiments. Different alphabets indicate statistically significant values, *p* < 0.05. Drug abbreviation: DMP, domperidone. Other abbreviation: DA, dopamine.

**Figure 8 ijms-24-04088-f008:**
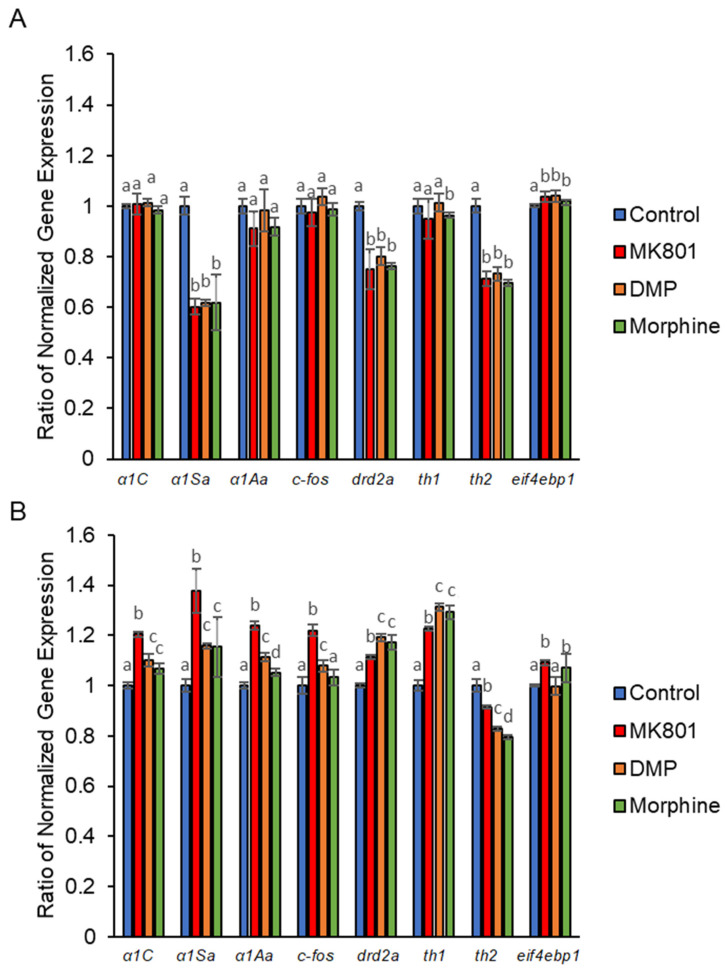
Altered gene expression in zebrafish larvae following exposure to MK801, DMP, and morphine. (**A**) Short-term effect on expression of *α1C* (F (3, 32) = 1.62, *p* = 0.91), *α1Sa* (F (3, 32) = 1.57, *p* = 0.22), *α1Aa* (F (3,32) = 1.99, *p* = 0.13), *c-fos* (F (3, 32) = 1.50, *p* = 0.67), *drd2a* (F (3, 32) = 53.34, *p* = 0.00), *th1* (F (3, 32) = 6.18, *p* = 0.02), *th2* (F (3, 32) = 283.76, *p* = 0.00), and *eif4ebp1* (F (3, 32) = 25.18, *p* = 0.00). (**B**) Long-term effect on expression of *α1C* (F (3,32) = 7.06, *p* = 0.00), *α1Sa* (F (3, 32) = 15.60, *p* = 0.00), *α1Aa* (F (3, 32) = 25.45, *p* = 0.00), *c-fos* (F (3,32) = 10.22, *p* = 0.01), *drd2a* (F (3, 32) = 264.97, *p* = 0.00), *th1* (F (3, 32) = 292.13, *p* = 0.00), *th2* (F (3, 32) = 617.59, *p* = 0.00), and *eif4ebp1* (F (3,32) = 20.76, *p* = 0.00). (**C**) Effect of morphine withdrawal of short-term morphine treatment on *α1C* (t (16) = 0.19, *p* = 0.66), *α1Sa* (t (16) = 1.13, *p* = 0.06), *α1Aa* (t (16) = 3.84, *p* = 0.10), *c-fos* (t (16) = 6.54, *p* = 0.02), *drd2a* (t (16) = 42.67, *p* = 0.00), *th1* (t (16) = 54.81, *p* = 0.00), *th2* (t (16) = 0.81, *p* = 0.38), and *eif4ebp1* (t (16) = 11.33, *p* = 0.00). (**D**) Effect of morphine withdrawal after long-term morphine treatment on *α1C* (t (16) = 0.69, *p* = 0.42), *α1Sa* (t (16) = 2.97, *p* = 0.10), *α1Aa* (t (16) = 24.37, *p* = 0.00), *c-fos* (t (16) = 7.49, *p* = 0.01), *drd2a* (t (16) = 18.02, *p* = 0.00), *th1* (t (16) = 18.11, *p* = 0.01), *th2* (t (16) = 28.78, *p* = 0.00), and *eif4ebp1* (t (16) = 8.10, *p* = 0.01). Data expressed as mean ± SEM, *n* = 50 of 3 independent experiments. Different alphabets indicate statistically significant values, *p* < 0.05. Drug abbreviation: DMP, domperidone.

**Table 1 ijms-24-04088-t001:** Primer sequences of candidate genes.

Protein	Gene	Primer Sequences (5′ → 3′)	GenBank Accession No	References
Ca_v_1.2	*α1C*	F: ACGGAGTCACTCTCCGACAC	XM_009300335	[120]
		R: AGAGAGGGCACAGGCTGATA		
Ca_v_1.1a	*α1Sa*	F: TCTATAGGCGTGCTGGAGGT	NM_001146150	[120]
		R: GCTATCTGCGAGCTGTAGGG		
Ca_v_2.1a	*α1Aa*	F: TGCTACCCAGCCACATGATA	ENSDARG00000037905	[120]
		R: TGGTAGAGAGTGAGGGTAAA		
Dopamine receptor D2a	*drd2a*	F: TGGTACTCCGGAAAAGACG	NM_183068	[121]
		R: ACTCGGGATGGGTGCATTTC		
c-FOS	*c-fos*	F:GCAGAGCATTGGCAGGAG	DQ003339	[122]
		R: CCCTTCGGATTCTCCTTTTCT		
Tyrosine hydroxylase 1	*th1*	F: GACGGAAGATGATCGGAGACA	XM_682702.1	[123]
		R: CCGCCATGTTCCGATTTCT		
Tyrosine hydroxylase 2	*th2*	F: CTCCAGAAGAGAATGCCACATG	NM_001001829.1	[124]
		R: ACGTTCACTCTCCAGCTGAGTG		
Eukaryotic translation initiation factor 4E-binding protein	*eif4ebp1*	F: AACGACAAGGTGCAAAGAC	NM_199645	[125]
		R: GTGGTTGGAATTGCCTGACT		
Beta actin (β-actin)	*actb1*	F: AAGCTGTGACCCACCTCACG	AF057040	[126]
		R:GGCTTTGCACATACCGGAGC		

## Data Availability

Not applicable.

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
