# Peer review of "Integrative Roles of Dopamine Pathway and Calcium Channels Reveal a Link between Schizophrenia and Opioid Use Disorder"

_ijms, 2023, doi:10.3390/ijms24044088_

Round 1

Reviewer 1 Report

In the manuscript “ Integrative Roles of Dopamine Pathway and Calcium Channels 2 Reveal a Link between Schizophrenia and Opioid Use Disorder” authors describe the effect of domperidone (antipsychotic - dopamine D2 receptor antagonist) and morphine in comparison with MK-801 (NMDA receptor antagonist) on zebrafish larvae with respect to behavior, dopaminergic neurons and expression of genes related to Schizophrenia. Overall, the manuscript is drafted very well but I have several concerns that need to be addressed before considering it for publication.

1.       Assay used for the locomotor activity is not a standardized assay. The manual observation and scoring could introduce bias and error. Recording larvae over a period and analyzing several parameters such as swimming speed, distance moved, hyper and hypo activity phases etc. are more acceptable in context of zebrafish larval locomotor activity. This can be performed by recording multiple larvae (in separate arena/dishes) with cheap camera (or cell phone camera) and analyzing videos with ImageJ based plug in or any other open-source programs.

2.       The social interaction assay also involved manual scoring, hence would be subject to possibility of bias and error. The assay is conducted at 10 days post fertilization, this is too early for larvae to show social interaction (please refer to  - Dreosti Elena, Frontiers in Neural Circuits, 2015). If authors could show this social interaction conclusively using the recording and presenting the tracks then it would be of really high impact.

 For the above-mentioned reasons, the data and conclusions from these experiments are not very convincing.

 3.       The data presented in Fig.3 and 4 is interesting but it is not clear if the quantitation was carried out using the ImageJ plugin or its manual counting. Such experiments require critical analysis, please refer to Liu et.al Plos One 2016 to see the acceptable standard for such analysis.  

Minor concerns

Use of color bar graphs and conventional asterisks symbols to show significance would be better.

Author Response

Assay used for the locomotor activity is not a standardized assay. The manual observation and scoring could introduce bias and error. Recording larvae over a period and analyzing several parameters such as swimming speed, distance moved, hyper and hypo activity phases etc. are more acceptable in context of zebrafish larval locomotor activity. This can be performed by recording multiple larvae (in separate arena/dishes) with cheap camera (or cell phone camera) and analyzing videos with ImageJ based plug in or any other open-source programs.

The locomotor assay used in this study was adapted from the previously validated work (Boehmler et al., 2007; Sterling et al., 2016), whereby the number of times a zebrafish crossing a defined line was taken as the measure for locomotor activity. 

The above work has been cited in Section 4.3 and added to References.

The social interaction assay also involved manual scoring, hence would be subject to possibility of bias and error. The assay is conducted at 10 days post fertilization, this is too early for larvae to show social interaction (please refer to  - Dreosti Elena, Frontiers in Neural Circuits, 2015). If authors could show this social interaction conclusively using the recording and presenting the tracks then it would be of really high impact.

Tracked trajectories have been included as supplementary results.

  The data presented in Fig.3 and 4 is interesting but it is not clear if the quantitation was carried out using the ImageJ plugin or its manual counting. Such experiments require critical analysis, please refer to Liu et.al Plos One 2016 to see the acceptable standard for such analysis.

The quantitation of data was based on ImageJ plugin. Manual counting was only used for verification. Therefore, description of manual counting was removed to avoid confusion.

Use of color bar graphs and conventional asterisks symbols to show significance would be better.

Colour bar graphs have been used. Alphabets were remained instead of asterisks to indicate the significance difference between each treatment based on post hoc Tukey’s test.

Reviewer 2 Report

The article is original and very interesting providing evidence
that there is an association between opioid addiction and schizophrenia.

I suggest few editing corrections.

1. lines 22-24- font seems to be bigger than the rest of the abstract

2 line 253 neurophathic- correct in neuropathic

3. In References section, after my knowledge the Doi code of the articles is required. Recheck the Instructions for Authors

Congrats for your work!

Author Response

Lines 22-24- font seems to be bigger than the rest of the abstract

The font size for lines 22-24 has been rectified.

Line 253 neurophathic- correct in neuropathic

The typo has been corrected.

In References section, after my knowledge the Doi code of the articles is required. Recheck the Instructions for Authors

The doi codes have been added to the references.

Reviewer 3 Report

Review for the manuscript Integrative Roles of Dopamine Pathway and Calcium Channels 2 Reveal a Link between Schizophrenia and Opioid Use Disorder submitted to IJMS.

 Dear Editor, thank you for the opportunity to review this manuscript. After careful evaluation, I suggest modifications before it can be accepted for publication in IJMS.

Overall comments: This is a very interesting and well-conducted study using a zebrafish model to investigate the differential effects of morphine and domperidone on dopaminergic neurons leading to alterations in gene expression, dopamine synthesis, and locomotor and social behaviors.

ABSTRACT

            The abstract is fine. However, I do not think that the use of italics for “α1C, α1Sa, α1Aa, drd2a, and th1” is necessary. See other italics used in the abstract. I suggest removing this format.

KEYWORDS

            I suggest including domperidone and morphine in the keywords.

INTRODUCTION

            This section is rich with much relevant information. However, I miss the inclusion of newer references. I suggest including references published in 2021 and 2022. As examples, the authors can check very good studies in PUBMED, such as:

Strømme MF, Mellesdal LS, Bartz-Johannesen CA, Kroken RA, Krogenes ML, Mehlum L, Johnsen E. Use of Benzodiazepines and Antipsychotic Drugs Are Inversely Associated With Acute Readmission Risk in Schizophrenia. J Clin Psychopharmacol. 2022 Jan-Feb 01;42(1):37-42. doi: 10.1097/JCP.0000000000001497. PMID: 34928559; PMCID: PMC8677602.

Strømme MF, Mellesdal LS, Bartz-Johannesen C, Kroken RA, Krogenes M, Mehlum L, Johnsen E. Mortality and non-use of antipsychotic drugs after acute admission in schizophrenia: A prospective total-cohort study. Schizophr Res. 2021 Sep;235:29-35. doi: 10.1016/j.schres.2021.07.009. Epub 2021 Jul 21. PMID: 34303258.

These above references can help introducing about ilicit drugs and schizophrenia.

Manion MTC, Glasper ER, Wang KH. A sex difference in mouse dopaminergic projections from the midbrain to basolateral amygdala. Biol Sex Differ. 2022 Dec 30;13(1):75. doi: 10.1186/s13293-022-00486-4. PMID: 36585727; PMCID: PMC9801632.

Panayi MC, Boerner T, Jahans-Price T, Huber A, Sprengel R, Gilmour G, Sanderson DJ, Harrison PJ, Walton ME, Bannerman DM. Glutamatergic dysfunction leads to a hyper-dopaminergic phenotype through deficits in short-term habituation: a mechanism for aberrant salience. Mol Psychiatry. 2022 Dec 2. doi: 10.1038/s41380-022-01861-8. Epub ahead of print. PMID: 36460723.

            These two above references can help the Introduction of Dopaminergic dysfunction and schizophrenia.

METHODS

            This section was adequately described.

Domperidone (DMP) was defined in line 101. It is not necessary to define again in line 320.

RESULTS

            I do not see a reason to keep this sentence in lines 105-106: “Firstly, we tested the locomotor behavior of zebrafish larvae upon exposure to MK-, DMP and morphine.” This is more like the Methods section.

            Please, define DMP in the legend of Figure 1. The same is true for the other Figures.

            Figure 6 is confusing. I suggest that authors choose a better illustration in the histogram bars in order to differentiate one treatment from another. As it stands, it’s very difficult to visualize. Moreover, this Figure’s legend appears before C) and D). Please, put it at the end of the Figure.

DISCUSSION

            This section is adequate. I have only one suggestion:

While Figure 7 is fine, I don’t see the need to keep a Figure in the Discussion. Also, if authors decide to leave this Figure and it was based on reference 107, it is necessary to refer to these authors in the figure caption.

CONCLUSION

            In this section, the authors say that "In summary, our study has integrated behavioral, biochemical, gene expression, and confocal microscopy data in drug-treated zebrafish larvae, to provide evidence that there is an association between opioid addiction and schizophrenia." From the way it is written, the relevant results obtained in this study are not clear. I suggest changing it to a sentence that clarifies the main findings of the study and not just saying that the study was designed to "provide evidence". Was the evidence found?

REFERENCES

        As pointed out above, I suggest including newer references in the Introduction section.

Author Response

The abstract is fine. However, I do not think that the use of italics for “α1C, α1Sa, α1Aa, drd2a, and th1” is necessary. See other italics used in the abstract. I suggest removing this format.

The italic format in the Abstract has been removed.

I suggest including domperidone and morphine in the keywords.

Domperidone and morphine have been added to the list of keywords.

 This section is rich with much relevant information. However, I miss the inclusion of newer references. I suggest including references published in 2021 and 2022. As examples, the authors can check very good studies in PUBMED, such as:

Strømme MF, Mellesdal LS, Bartz-Johannesen CA, Kroken RA, Krogenes ML, Mehlum L, Johnsen E. Use of Benzodiazepines and Antipsychotic Drugs Are Inversely Associated With Acute Readmission Risk in Schizophrenia. J Clin Psychopharmacol. 2022 Jan-Feb 01;42(1):37-42. doi: 10.1097/JCP.0000000000001497. PMID: 34928559; PMCID: PMC8677602.

Strømme MF, Mellesdal LS, Bartz-Johannesen C, Kroken RA, Krogenes M, Mehlum L, Johnsen E. Mortality and non-use of antipsychotic drugs after acute admission in schizophrenia: A prospective total-cohort study. Schizophr Res. 2021 Sep;235:29-35. doi: 10.1016/j.schres.2021.07.009. Epub 2021 Jul 21. PMID: 34303258.

These above references can help introducing about ilicit drugs and schizophrenia.

Manion MTC, Glasper ER, Wang KH. A sex difference in mouse dopaminergic projections from the midbrain to basolateral amygdala. Biol Sex Differ. 2022 Dec 30;13(1):75. doi: 10.1186/s13293-022-00486-4. PMID: 36585727; PMCID: PMC9801632.

Panayi MC, Boerner T, Jahans-Price T, Huber A, Sprengel R, Gilmour G, Sanderson DJ, Harrison PJ, Walton ME, Bannerman DM. Glutamatergic dysfunction leads to a hyper-dopaminergic phenotype through deficits in short-term habituation: a mechanism for aberrant salience. Mol Psychiatry. 2022 Dec 2. doi: 10.1038/s41380-022-01861-8. Epub ahead of print. PMID: 36460723.

             These two above references can help the Introduction of Dopaminergic dysfunction and schizophrenia.

Recent references have been included as suggested.

Domperidone (DMP) was defined in line 101. It is not necessary to define again in line 320.

The definition of domperidone has been removed from Section 4.2.

I do not see a reason to keep this sentence in lines 105-106: “Firstly, we tested the locomotor behavior of zebrafish larvae upon exposure to MK-, DMP and morphine.” This is more like the Methods section.

This sentence has been removed.

 Please, define DMP in the legend of Figure 1. The same is true for the other Figures.

The definitions for DMP and other relevant abbreviations have been added to the captions of all figures.

Figure 6 is confusing. I suggest that authors choose a better illustration in the histogram bars in order to differentiate one treatment from another. As it stands, it’s very difficult to visualize. Moreover, this Figure’s legend appears before C) and D). Please, put it at the end of the Figure.

The histograms have been modified. The figure legend has been moved to the end of Figure 6.

While Figure 7 is fine, I don’t see the need to keep a Figure in the Discussion. Also, if authors decide to leave this Figure and it was based on reference 107, it is necessary to refer to these authors in the figure caption.

Figure 7 has been removed.

In this section, the authors say that "In summary, our study has integrated behavioral, biochemical, gene expression, and confocal microscopy data in drug-treated zebrafish larvae, to provide evidence that there is an association between opioid addiction and schizophrenia." From the way it is written, the relevant results obtained in this study are not clear. I suggest changing it to a sentence that clarifies the main findings of the study and not just saying that the study was designed to "provide evidence". Was the evidence found?

The sentence has been rephrased.

 As pointed out above, I suggest including newer references in the Introduction section.

References have been updated.

Round 2

Reviewer 1 Report

Authors have attempted to address my concerns about manual scoring of the behavior. Authors have cited Boehmler et al., 2007 in which same assay was used. The second study cited from Sterling et al., 2016 used the similar assay but the behavior recording was analyzed for locomotion across the grid. There has been lot of technological advancement and availability of better tools (commercial and open source) since these studies were published and the current standard in the field is much higher. In the situation where manual scoring becomes the only option, then it should be carried out as double-blind study. Authors have used ToxTrac for tracking larvae in social behavior assay which become more convincing observation, this should be used throughout the study to extract relevant information (velocity, distance moved, period of hyper/hypo activity, freezing etc.) for locomotor behavior as well.

The supplementary figure should be moved to main figure, it’s a good data to be included as main figure.

Author Response

. Comment

Response

1

Authors have attempted to address my concerns about manual scoring of the behavior. Authors have cited Boehmler et al., 2007 in which same assay was used. The second study cited from Sterling et al., 2016 used the similar assay but the behavior recording was analyzed for locomotion across the grid. There has been lot of technological advancement and availability of better tools (commercial and open source) since these studies were published and the current standard in the field is much higher. In the situation where manual scoring becomes the only option, then it should be carried out as double-blind study. Authors have used ToxTrac for tracking larvae in social behavior assay which become more convincing observation, this should be used throughout the study to extract relevant information (velocity, distance moved, period of hyper/hypo activity, freezing etc.) for locomotor behavior as well.

Total distance travelled has been included (as Figure 2).

2

The supplementary figure should be moved to main figure, it’s a good data to be included as main figure

Figure S1 has been renamed as Figure 4